# Perception of breast cancer risk factors: Dysregulation of TGF-β/miRNA axis in Pakistani females

Fayyaz Ahmed[1], Muhammad Adnan🔘[1], Ayesha Malik[1], Somayya Tariq[1], Farukh Kamal[2], Bushra Ijaz🔘[1]*

1 Laboratory of Applied and Functional Genomics, National Center of Excellence in Molecular Biology, University of the Punjab Lahore, Lahore, Pakistan, 2 Department of Pathology, Fatima Jinnah Medical University, Lahore, Pakistan

* bijaz@cemb.edu.pk

**Data Availability Statement:** All relevant data are within the manuscript and its Supporting information files.

**Funding:** The authors received no specific funding for this work.

## Abstract

Breast cancer poses a serious health risk for women throughout the world. Among the Asian population, Pakistani women have the highest risk of developing breast cancer. One out of nine women is diagnosed with breast cancer in Pakistan. The etiology and the risk factor leading to breast cancer are largely unknown. In the current study the risk factors that are most pertinent to the Pakistani population, the etiology, molecular mechanisms of tumor progression, and therapeutic targets of breast cancer are studied. A correlative, cross-sectional, descriptive, and questionnaire-based study was designed to predict the risk factors in breast cancer patients. Invasive Ductal Carcinoma (90%) and grade-II tumor (73.2%) formation are more common in our patient's data set. Clinical parameters such as mean age of 47.5 years (SD ± 11.17), disturbed menstrual cycle (> 2), cousin marriages (repeated), and lactation period (< 0.5 Y) along with stress, dietary and environmental factors have an essential role in the development of breast cancer. In addition to this *in silico* analysis was performed to screen the miRNA regulating the TGF-beta pathway using TargetScanHuman, and correlation was depicted through Mindjet Manager. The information thus obtained was observed in breast cancer clinical samples both in peripheral blood mononuclear cells, and biopsy through quantitative real-time PCR. There was a significant dysregulation (**P>0.001) of the *TGF-β1* signaling pathway and the miRNAs (miR-29a, miR-140, and miR-148a) in patients' biopsy in grade and stage specifically, correlated with expression in blood samples. miRNAs (miR-29a and miR-140, miR-148a) can be an effective diagnostic and prognostic marker as they regulate *SMAD4* and *SMAD2* expression respectively in breast cancer blood and biopsy samples. Therefore, proactive therapeutic strategies can be devised considering negatively regulated cascade genes and amalgamated miRNAs to control breast cancer better.

**Competing interests:** The authors have declared that no competing interests exist.

## Introduction

Breast cancer is a serious health risk for women throughout the world. According to World Health Organization (WHO), more than 1.7 million cases of breast cancer were diagnosed out of 14.1 million total cancer cases in the world. It is the highest occurring cancer in females, and its mortality rate is on the fifth number [1]. Among the Asian population, Pakistani women have the highest risk of developing breast cancer [2]. The estimated rate of occurrence in Pakistan is 1 in every 9 women [3]. Around 90,000 cases of breast cancer are reported annually in Pakistan, and approximately 40,000 deaths are caused by it. In 2015, among all cancer cases reported in Shaukat Khanum Memorial Cancer Hospital and Research Center (SKMCH&RC), Lahore, 22.03% were of breast cancer [4].

Although breast cancer is a heterogeneous disease, age is a common risk factor. As a woman grows older, her risk of developing breast cancer increases due to genetic and hormonal factors [5]. In Pakistan, most of the patients diagnosed with breast cancer are <40 years old and mostly presented at an advanced clinical stage [6]. This is an alarming threat to young females. Furthermore, several Asian population-based studies have revealed that the incidence of breast cancer is more pronounced at the age of 40 to 59 years. Mainly, a substantial breast cancer occurrence rate has been observed between 40–49, 50–59, 60–64, and 45–54 age in Korea, India, Sri Lanka, and Japan, respectively [7–10]. Other risk factors or causes of occurrence include the early age of onset of menstruation or late menopause. Women who do not have children or have their first child after the age of 30 are also at increased risk [5]. Other than non-genetic factors, many genetic causes also contribute to the etiology of breast cancer [11]. For instance, familial susceptibility to breast cancer accounts for approximately 10% of all breast cancer cases. Among the breast cancer genes, *BRCA1* and *BRCA2* are noteworthy [12]. Alterations in these genes are associated with a significantly increased risk of breast cancer [11,13,14].

The high prevalence of breast cancer has several associated reasons. One of them is late diagnosis. The women appear for diagnosis at an advanced stage, which affects the prognosis and treatment, thus limiting the survival chances [15]. However, there is a lack of data on pathophysiological features of breast cancer among Pakistani women, primarily due to the non-availability of screening centers. A mammogram is an effective imaging technique to detect breast cancer, but it is costly [16]. The lack of inexpensive facilities poses adverse effects on the early diagnosis and prognosis of breast cancer. Moreover, there is a need to properly evaluate patient's risk factors that are most pertinent in one population and focus on etiology, molecular mechanisms of tumor progression, and therapeutic targets of breast cancer.

Differential expression of genes in normal and tumor cells can help to understand the mechanism of disease development and progression [17,18]. Another group of regulatory entities involves microRNAs (miRNAs). These non-coding RNAs regulate the production of proteins from mRNA (messenger RNA) either through the repression of translation or mRNA degradation [19]. The role of miRNAs in cancer pathobiology is an emerging topic, and it can give insights to understand new ways to study the disease progression and therapy [17].

Most of the changes that occur in the tumor are either the genetic mutations or the deregulation of growth or developmental pathways (*Wnt*, Notch, hedgehog, etc.), which leads to uncontrolled growth of cells and thus results in cancer [20–22]. One such pathway is the Transforming Growth Factor-β1 (*TGF-β1*) pathway [23]. *TGF-β1* belongs to the family of cytokines that regulate the growth and differentiation of the cells. *TGF-β1* has been reported to stimulate apoptosis by preventing the epithelial cell cycle and thus shows the tumor suppressor effects [23]. Simultaneously, *TGF-β1* has also been reported to induce tumor growth and metastasis by stimulating the epithelial to mesenchymal transition (EMT) [24]. *TGF-β1* protein

(MW 25kDa) has a homodimeric structure, and it is released into the extracellular matrix (ECM) in the form of an inactive precursor. After it binds to membrane-bound receptor *TGF-β* receptor II (*TβRII*), which then induces *TβRI*. The activation of *TβRI* then assigns the intra-cellular receptor-regulated signaling proteins: R-Smads which include *SMAD2* and *SMAD3* [25]. The formation of this receptor and R-Smad complex and *SMAD4* (co-*SMAD*) binds to this complex to form a trimeric complex of *SMADs*. The trimeric complex translocases to the nucleus where it controls the activity of the TGF-β1 dependent target genes [25]. The signaling of *TGF-β1* is controlled by the negative feedback mechanism that involves I-Smads (*SMAD6* and *SMAD7*) [26]. I-SMADs, or inhibitory SMADs, make a complex with activated *TβRI* and restrict the cellular response, thus inhibiting the *TGF-β1* activity. The level of *TGF-β1* is found to increase in breast cancer, and its level can be related to the grade and stage of cancer and the response of tumor growth to chemotherapy.

In the present study, the demographic data and risks of breast cancer development pertaining to the Pakistani population were evaluated. In addition, we evaluated the expression of the *TGF-β1* pathway in normal and tumor samples. The expression was correlated with the stage and grade of the patients not only in breast cancer biopsy samples but also in the blood samples. *TGF-β1* pathway deregulation has been reported in many cancers and other diseases [27,28]. Still, its expression in breast cancer has not been reported in the Pakistani population. Moreover, the expression was also checked in pre- and post-chemo treated patients' samples to observe the effect of chemotherapeutics on the regulation of the *TGF-β1* pathway. Additionally, this study also focuses on the expression analysis of miR-140, miR-148, and miR-29a in breast cancer clinical samples. These miRNAs target the *TGF-β1* pathway, and their expression analysis can help to use them as a therapeutic and diagnostic target. This analysis will further help ascertain the population-based diagnostic and therapeutic targets compared to expression profiles of *TGF-β1* pathway genes and miRNAs. The expression profiles would be helpful in early diagnosis, prognosis, and determining the grade, stage, and pre-and post-therapy-specific expressions of tumor cells.

## Methodology

### Design of study

To examine the intervening role of risk factors associated with breast cancer among the Pakistani population, a descriptive cross-sectional study was designed using self- or interview-administered validated questionnaires. Correlation of expression profile of blood and biopsy samples from cancer patients was conducted.

### Data collection

The study was approved by the CEMB, institutional review Committee, the Institutional Review Board (IRB), Sir Ganga Ram Hospital, and the IRB of Sheikh Zayed Hospital, Lahore, Pakistan. Medical history and relevant data were collected from 400 patients suffering from breast cancer in different hospitals of Pakistan, including Fatima Jinnah Medical College, Sir Ganga Ram Hospital, Sheikh Zayed Hospital. The written informed signed consent to participate in the study was taken by the patients. The questionnaire (S1 and S2 Files) encompassed different parameters to record the current and previous health status of patients. Therein, time of initial diagnosis before surgery, age, menstrual cycles per month, marital status, marriage in the family, number of children, lactation period, family and personal history of breast cancer, history of smoking, socio-economic status, stress, lifestyle, the dietary status of types of diet, and place of residence or locality were the included parameters.

## miRNA-target genes network analysis

The computational network analysis of miR-29a, miR-140, and miR148a was carried out to find possible target genes via TargetScanHuman v7.2 (http://www.targetscan.org/vert_72/) and miRDB (http://www.mirdb.org/). Further, the total number of predicted targets was assessed through MS Excel analysis tools and represented in the form of a Venn diagram. However, the correlation was depicted using Mindjet Manager 2019 (https://www.mindmanager.com/en/support/download-library/) in a network format. Furthermore, validated targets were determined through recent research investigations.

## Clinical sample collection

To study *TGFβ* signaling pathway, we collected 109 blood and fresh biopsy samples (1.5 cm) from the breast cancer patients undergoing Modified Radical Mastectomy (MRM) right after their surgical procedure. Immediately after collection, fresh biopsy samples were immersed in RNA Later solution (Ambion, USA). Inclusion criteria revolved around patients who underwent MRM having Invasive Ductal Carcinoma (IDC) with tumor grade II and III. Non-malignant tissue was employed as control which was taken from the tumor-free marginal area or its borderline/para-neoplastic junction area or the distant location from the tumor area. Normal tissue may be connective and stromal/glandular tissue of breast biopsy. The status of normal biopsy was decided by the consensus of surgeons, pathologists, and molecular biologists. Benign tumor including fibro-adenoma was also used as a control in our study.

## RNA extraction from biopsy and blood samples

RNA was extracted by using the optimized TRIzol method [29]. For RNA extraction, 50 mg of the tissue was ground in liquid nitrogen and followed by the standard TRIzol method. In blood samples, peripheral blood mononuclear cells (PBMCs) were isolated from the whole blood. For this purpose, the blood was taken in a sterile culture tube, and Erythrocyte Lysis (EL) Buffer (Qiagen) was added into it in 1:3 v/v (3 ml blood: 9 ml EL buffer). Further, the obtained white pellet of PMBCs was processed for RNA isolation.

## Synthesis of miRNA cDNA

The cDNA of miRNA was synthesized from total RNA using Poly A polymerase (Sigma-Aldrich®). 10μl of reaction volume was prepared containing 100ng of RNA, universal RT-primer (CAGGTCCAGTTTTTTTTTTTTTTTTAC), Poly A polymerase buffer, Poly A polymerase, dATPs, dNTPs, and MMLV-RT. After adding the above components, the tube was spun shortly and incubated in thermal cycler at 42˚C for 60 minutes to synthesize cDNA and 95˚C for 5 minutes for enzyme inactivation. When the reaction was completed, the miRNA cDNA was stored at -20˚C.

## Quantitative RT-PCR for expression profiling

For gene and miRNA expression analysis, 1 μl of cDNA was added to components of the PCR mixture. The expression of *TGF-β1* pathway genes and miRNAs in different grades of breast cancer was quantified through conducting Quantitative Real-Time PCR (qRT-PCR) using ABI 7500 qRT-PCR system. Gene-specific and miRNA primers (S1 and S2 Tables) were used along with 2x SYBR Green Master Mix (Applied Biosystems™) according to the manufacturer's manual [30]. The GAPDH gene was used for normalization as an internal control for mRNA expression analysis and RNU6 for miRNA. The relative expression analysis was done by using SDS 3.1.1 software provided by ABI. Each reaction was performed in triplicate.

## Statistical analysis

Graph Pad Prism 5.0 software was used to illustrate results graphically. All statistical analyses, including one-way ANOVA, Post-hoc t-tests, descriptive analysis, and Pearson correlation analysis, were carried out in IBM SPSS Statistics (version 23) software.

## Results

### Demographic profiling of breast cancer patients

In this study, 400 breast cancer patients from Lahore, Pakistan, participated. Out of them, 299 breast cancer patients completed the questionnaire (75% response rate). The cohort includes only female patients with a mean age of 47.5 years (SD ± 11.17), with an average range of 20–80 years. Demographic profiling of breast cancer patients among the Pakistani population is shown in Table 1. Notably, 81% of the patients were married within their family, having tumor type IDC (90%), DCIS (8.6%), Edema, and ILC (0.7%). This indicated that invasive ductal carcinoma is more frequent in the Pakistani female population as compared to ductal carcinoma *in situ*. Moreover, 73.2% of patients had tumor grade II, 23.4% grade III, while 7 and 2.7% had lump and tumor grade I. Therefore, most of the patients in this study were tumor Grade II. Most of the female patients were housewives (84.3%), illiterate (32.8%), living in a rural environment (47.8%) under continuous stress conditions (91.6%), and frequent vegetarians (most of the time take vegetables but 2–3 times a month take in meat 52.5%) Table 1.

To explore genetic factors related to the disease, the family history of breast cancer patients was recorded. It was observed that only 4.59% of the patients had a family history of breast cancer while the remaining 95.40% of patients had no family history of breast cancer, signifying the possible involvement of other factors in disease causation. The numbers of children for each patient's family were also perceived to evaluate the possible role of pregnancy in regulating breast cancer augmentation. It has been observed that 3.4% of females showed no pregnancy, 47% females had 1–3 pregnancies, and 49.5% females had more than 3 pregnancies, indicating that the risk of breast cancer is indirectly linked with the number of pregnancies. The regulation of the menstrual cycle is another factor linked with the number of children and lactation period, as all three parameters are regulated by the female's sex hormones. Therefore, in the current study, the history of the menstrual cycle was also inquired. The results indicated that 31% of females had menstrual cycle 01/month, 30% had 02 cycles/month. In comparison, 39% of females had disturbed menstrual cycle, i.e., >02 cycles/month, showing that the breast cancer risk increased with deregulation of the menstrual cycle.

Pearson correlation analysis of the relationship between age, marriage, number of children, environment, stress condition, menstruation, diet, and occupation tumor type and tumor grade are shown in Table 2. As it has been shown that in breast cancer patients, mensuration had a significant negatively correlation with (r = -0.1, p ≤ 0.037), diet (r = -0.164, p = 0.005) and education (r = -0.168, p = 0.004) but positive significant relationship with stress factor (r = 0.146, p = 0.012). Marriage type (weather patients are married within family or out of family) showed a negative significant correlation with type of environment (r = -0.177, p = 0.042) and stress factor (r = -0.172, p = 0.003), while marriage type showed significant positive correlation with type of diet (r = 0.122, p = 0.035). Number of children had a highly significant positive relationship with age of patient (r = 0.239, p = 0.000). Diet had a positive significant correlation with education (r = 0.13, p = 0.024). Occupation had a positive highly significant relationship with education (r = 0.301, p = 0.000), while showed a significant negative relationship with stress (r = -0.154, p = 0.008). Education had a positive significant relationship with type of environment (r = 0.561, p = 0.000), whereas it has negative relationship with stress

**Table 1. Prevalence of different clinical parametres in breast cancer patients (n = 299).**

| Sr. No. | Clinical Parameters | | | Prevalence Percentage (%age) |
|---|---|---|---|---|
| 01 | Age | 20–30 Y | | 4.34 |
| | | 31–50 Y | | 61.87 |
| | | > 50Y | | 33.44 |
| | | Menstrual cycles/month (154)/78.57% | 01 | 31 |
| | | | 02 | 30 |
| | | | > 02 | 39 |
| | | Menopause (42)/21.42% | 45-55Y | 54.74 |
| | | | >55 | 45.23 |
| 02 | Marriage | Married within family | | 81 |
| | | Married out of family | | 19 |
| | | Divorced | | 0.0 |
| 03 | No. of Children | 0.0 | | 3.4 |
| | | 01–03 | | 47 |
| | | >03 | | 49.5 |
| 04 | Type of Tumor | Invasive Ductal Carcinoma (IDC) | | 90 |
| | | Invasive Lobular Carcinoma (ILC) | | 0.7 |
| | | Edema | | 0.7 |
| | | Ductal Carcinoma In-situ (DCIS) | | 8.6 |
| 05 | Tumor Grade | I | | 2.7 |
| | | II | | 73.2 |
| | | III | | 23.4 |
| | | Lump | | 0.7 |
| 06 | Stress Factors | Short term (acute) | | 8 |
| | | Continuous stress (chronic) | | 92 |
| 07 | Life Style | i) Occupation | Housewife | 84.3 |
| | | | Working (jobs) | 15.7 |
| | | ii) Diet | Vegetarian | 52.5 |
| | | | Meat | 25.4 |
| | | | Fruits | 22.1 |
| | | iii) Education | Illiterate | 33 |
| | | | Primary or above | 31 |
| | | | Middle or above | 25.0 |
| | | | Intermediate or above | 11.0 |
| | | iv) Socio-Economic Status | Lower class | 47.8 |
| | | | Average Class | 30.4 |
| | | | Upper Class | 21.7 |

(r = -0.194, p = 0.001). It is observed that age, marriage, number of children, environment, stress condition, menstruation, diet, and occupation tumor type and tumor grade are important risk factors in increasing the rate of breast cancer among Pakistani population.

## Predicted targets of miR-29a, miR-140, and miR-148a

To explicate the biological mechanisms behind miRNA regulation, predicted targets were identified. The analysis revealed 4855, 429, and 3853 predicted targets of miR-29a, miR-140, and miR-148a, respectively (Fig 1). However, 1784 targets were found common between miR-29a and miR-148a, while 200 in miR-29a and miR-140. In contrast, 174 common targets were

**Table 2. Pearson correlation analysis of the relationship between different risk factors associated with breast cancer.**

|  |  | Age | Menstruation | Marriage | No of Children | Tumor type | Tumor grade | Diet | Occupation | Education | Environment | Stress |
|---|---|---|---|---|---|---|---|---|---|---|---|---|
| **Age** | r | 1 | .074 | .101 | **.239**** | -.051 | -.088 | .008 | .040 | .096 | .044 | -.026 |
|  | p |  | .200 | .080 | .000 | .377 | .130 | .887 | .493 | .098 | .450 | .649 |
| **Menstruation** | r |  | 1 | -.1* | .028 | .057 | .004 | **.164**** | -.021 | **-.168**** | -.003 | **.146*** |
|  | p |  |  | .037 | .627 | .323 | .940 | .005 | .724 | .004 | .954 | .012 |
| **Marriage** | r |  |  | 1 | .096 | .080 | .003 | **.122*** | .067 | .036 | **-.117*** | **-.172**** |
|  | p |  |  |  | .097 | .166 | .953 | .035 | .248 | .536 | .042 | .003 |
| **No. of children** | r |  |  |  | 1 | -.070 | .040 | .024 | -.077 | -.025 | .032 | .076 |
|  | p |  |  |  |  | .227 | .490 | .685 | .185 | .664 | .588 | .193 |
| **Tumor type** | r |  |  |  |  | 1 | .048 | .044 | -.087 | -.038 | -.009 | -.050 |
|  | p |  |  |  |  |  | .407 | .450 | .133 | .515 | .876 | .384 |
| **Tumor grade** | r |  |  |  |  |  | 1 | .068 | .031 | -.040 | -.107 | .111 |
|  | p |  |  |  |  |  |  | .238 | .599 | .487 | .063 | .056 |
| **Diet** | r |  |  |  |  |  |  | 1 | .083 | **.130*** | -.103 | -.083 |
|  | p |  |  |  |  |  |  |  | .152 | .024 | .074 | .150 |
| **Occupation** | r |  |  |  |  |  |  |  | 1 | **.301**** | .052 | **-.154**** |
|  | p |  |  |  |  |  |  |  |  | .000 | .367 | .008 |
| **Education** | r |  |  |  |  |  |  |  |  | 1 | **.561**** | **-.194**** |
|  | p |  |  |  |  |  |  |  |  |  | .000 | .001 |
| **Environment** | r |  |  |  |  |  |  |  |  |  | 1 | .075 |
|  | p |  |  |  |  |  |  |  |  |  |  | .194 |
| **Stress** | r |  |  |  |  |  |  |  |  |  |  | 1 |
|  | p |  |  |  |  |  |  |  |  |  |  |  |

**. Correlation is significant at the 0.01 level (2-tailed).

*. Correlation is significant at the 0.05 level (2-tailed).

determined in miR-140 and miR-148a. Furthermore, all three miRNAs regulate 93 common targets.

## miRNA-target genes network analysis

The computational network analysis of miR-29a, miR-140, and miR148a predicted their interactions with several target genes, including *TGF-β1, FOXO3, MAPK, and IGFR* cascades and other intricated regulatory genes as illustrated in Figs 2 and 3. miR-29a shows a vast network of interacting genes cross-talking and mediating their function through *AKT, EGFR,* and *TGFβ* pathways. miR-29a predictably interact with *BAMBI, TGIF1, TGFβR2, SMAD2, TGFβR3, STAT1, SMAD9, EGFR, SKP1, SMURF1, CACUL1, THBS1, AKT1,* and *FBN1* gene (Fig 2a). However, its validated interaction was found with *SMAD4, TGFβ,* and *IGF1R*. The literature revealed that miR-29a has an inverse relation with these three genes in the tumor. Like miR-29a, miR-140's predicted many targets such as *FBN1, SMURF1, SMAD2, TGFBR1, VEGFA, IGF1R, BMP2, CREB1, BCL2L2, BCL2L1, MAPK1* (Fig 2b). Moreover, several investigations showed that the downregulation of miR-140, in turn, upregulates *EGFR, VEGFA, IGFBP5, SMAD3, and SMAD2*.

While the predicted targets of miR-148a were: *TGIF1, IGF2R, SKP1, SMAD2, TGFBR2, SMAD4, SMAD9, AKT2, IRS1, IGF1R, BAMBI, MAPK1, GAREM, CACUL1, SMAD5, ROCK1, FOXG1, FOXO3, BCL2, FBN1* (Fig 2c). Similarly, miR-148a showed the relation with *SMAD2, TGFβ2, IRS1, IGF1R, BCL2, ROCK1, and STAT3*.

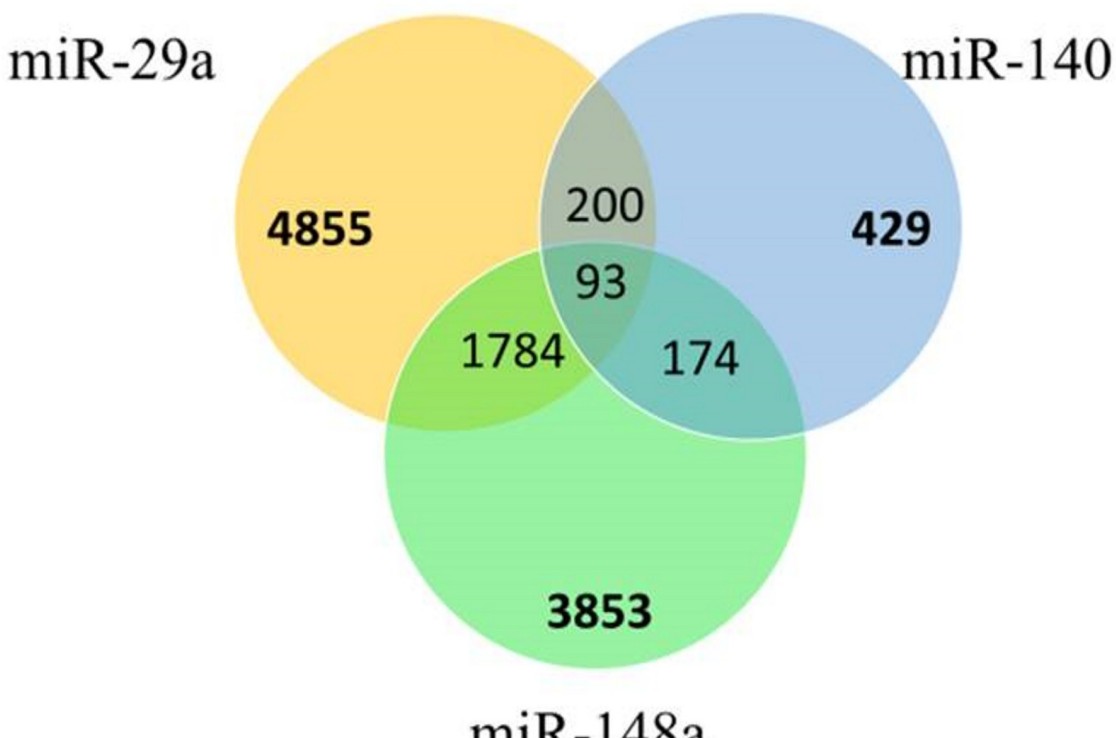

**Fig 1. Venn diagram displaying the number of differentially overlapping predicted target genes between miR-29a, miR-140, and miR-148a.**

### Expression analysis of *TGF-β1* pathway genes and miRNAs in breast cancer clinical samples

The expression of *TGF-β1* pathway genes and miRNAs was analyzed by qRT-PCR. To evaluate whether the expression was associated with the advanced grade of breast cancer, we analyzed the expression in pre-chemotherapy patients compared to control samples and post-chemotherapy patients. A separate comparison was done between blood and biopsy samples of grade II and grade III patients for correlation analysis and to predict biomarkers.

### *TGF-β* cascade expression analysis in grade-II and G-III breast cancer patients

The expression of *TGF-β1* (ligand) cascade genes was assessed in blood and biopsy samples of grade II and III breast cancer patients compared to control samples. The results revealed very low or almost no expression of *TGF-β1* in control samples. However, higher expression was detected in G-II and G-III samples. Relative quantitative analysis showed a 1-fold increased expression in both G-II and G-III pre-chemo clinical samples. However, patients who have undergone chemotherapy showed relatively differential expression patterns (Fig 4). Similar to *TGF-βI*, *TβRII* exhibited upsurge expression in G-II and G-III pre-chemo blood and biopsy samples. While, in post-chemo patient samples, a slightly raised level of expression was seen. Herein, 0.8-folds and 0.5-fold rise in *TβRII* expression was observed in pre-chemo blood samples of both GII and GIII, respectively. Whereas, in G-II pre-chemo biopsy samples, 1.7 times and GIII, 1.2-folds rise was detected. Contrary to this, no significant change was observed in

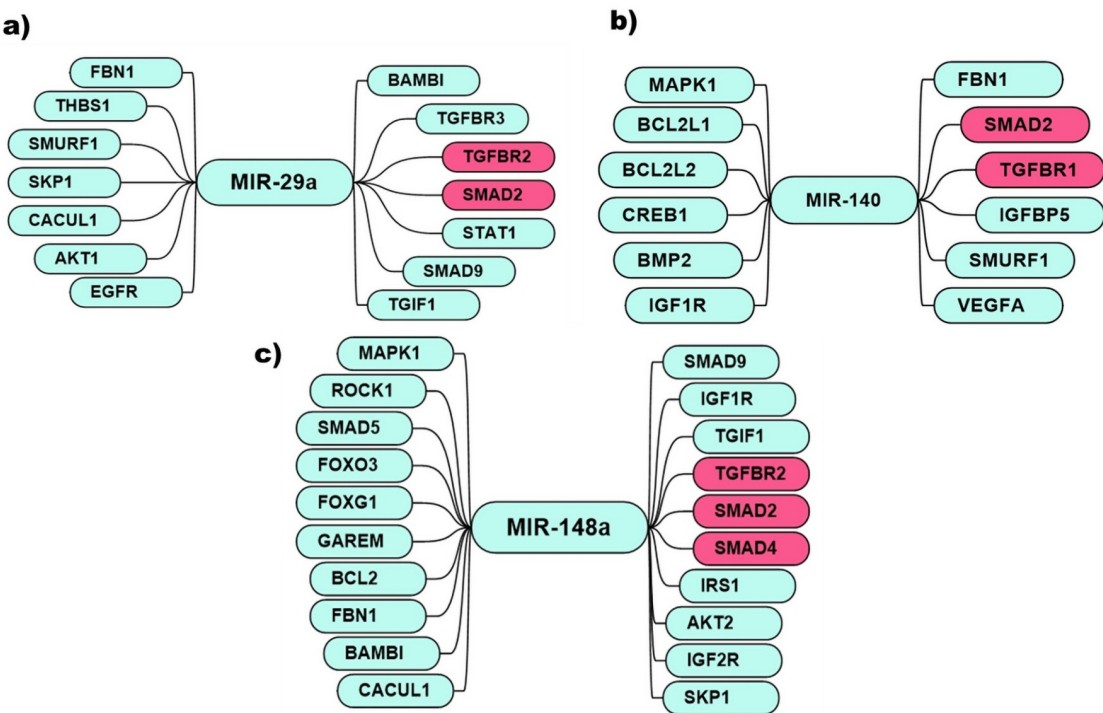

**Fig 2. miRNA-target gene network analysis.** Predicted targets of **a)** miR-29a, **b)** miR-140 **c)** miR-148a. Genes represented in the red box are studied in the current study in clinical samples.

post-chemotherapy patients in comparison to control samples. On the other hand, the expression level of *SMAD2* was seen to be increased approximately by 0.5 folds in G-II pre-chemo patients compared to control samples. Consequently, the mRNA expression of *SMAD2* in G-III pre-chemo blood samples was 0.6-fold higher than control samples, while it decreased to 0.7 folds in post-chemo samples (Fig 4). Contrary to this, *SMAD2* expression was elevated up to 0.6 folds in G-II pre-chemo biopsy samples compared to control. In contrast, no significant expression change was detected after chemotherapy. Subsequently, in G-III biopsy samples, *SMAD2* expression was increased up to 0.4-fold in pre-chemo compared to control samples; however, the expression decreased to 0.6 folds in post-chemo patients. *SMAD4* showed significantly high expression about 0.5–1 folds in blood samples of G-II and G-III patients, respectively. In comparison, control samples exhibited very low expression levels. Moreover, a 1–1.5-fold increase was seen in pre-chemotherapy patients of G-II and G-III biopsy samples (Fig 4b and 4d). Post-chemotherapy samples represented very small changes compared to their controls. SMAD7, an inhibitor of *TGF-β1* signaling, revealed approximately 0.3-fold escalated expression in G-II and G-III blood samples. In biopsy samples, a 0.8-fold rise was seen in pre-chemotherapy patients of G-II, while no change was present in post-chemo patients. However, In G-III, a 1-fold increase in pre-chemo patients while no considerable increase was observed in post-chemo patients except for the biopsy samples of G-II patients, where a reduction was noticed. p21 is also regulated by the *TGF-β1* pathway. It showed a higher expression in both G-II (0.6-fold) and G-III (0.4-fold) blood samples relative to normal. In comparison, expression was up-regulated to 1-fold in G-II and G-III pre-chemo biopsies. In post-chemo patients, no significant change of expression was observed compared to control samples.

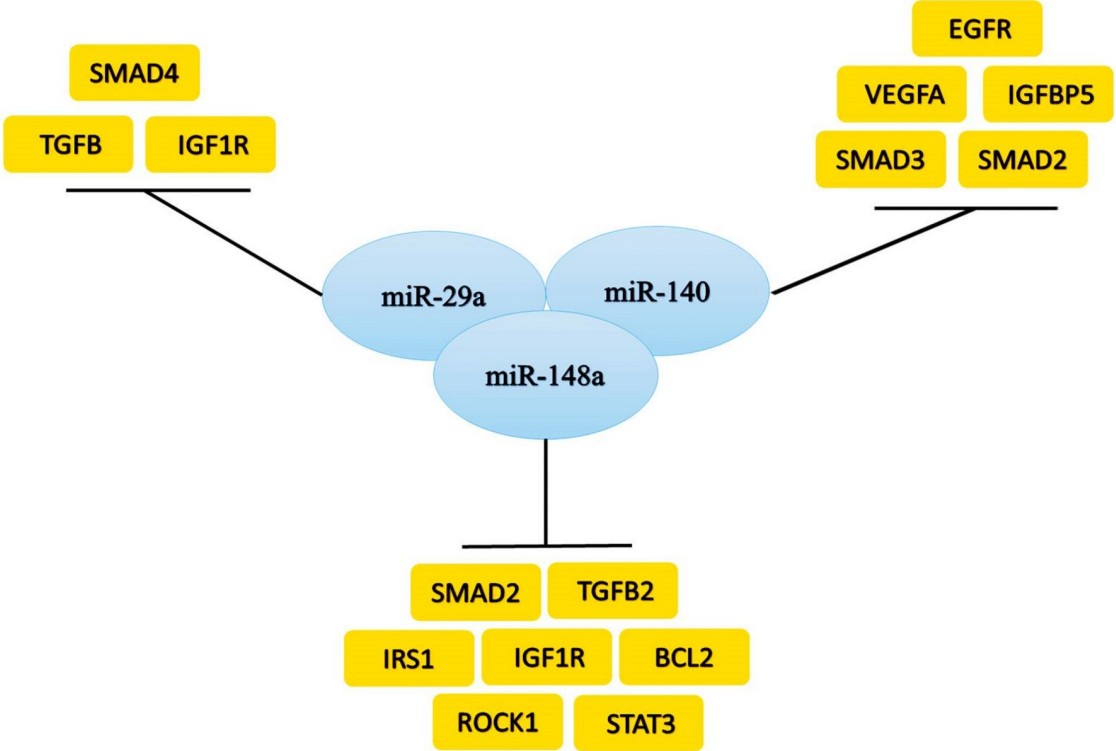

**Fig 3. miRNA validated interaction with genes. a)** miR-29a, **b)** miR-140 **c)** miR-148a inhibit the expression of the genes.

### Expression analysis of miR-29a, miR-140, and miR-148a in G-II and G-III breast cancer patients

The relative expression analysis of miR-29a using qRT-PCR showed significant differential expression in G-II and G-III blood and biopsy samples with a 1 to 1.2-fold reduction compared to control (Fig 5). In contrast, miR-140 showed 1-fold down-regulated expression in G-II and G-III blood and biopsy samples. There was no significant change in control and post-chemo samples. The expression level of miR-148a depicts the level of *SMAD2*. Real-time analysis showed 1 to the 1.4-fold reduced level of miR-148a in G-II and G-III pre-chemo blood and biopsy samples, respectively, compared to normal samples (Fig 5a–5d). While, after chemotherapeutic treatment of patients, miR-148a expression upsurges to similar levels as in normal patients.

### Interaction of SMAD2 and miR-140 and miR-148a

*SMAD2* gene, miR-140, and miR-148a expression patterns were found correlating between blood and biopsy samples of breast cancer patients. A relatively high expression of the *SMAD2* gene was detected in pre-chemotherapy patients of both GII and GIII blood and biopsy samples compared to control. In contrast to *SMAD2*, low expression of miR-140 and miR-148a was observed. This indicates an inverse relation between them (Fig 6a). Moreover, the expression of both miRNAs reverses to normal in post-chemotherapy patients' samples in both grades. *SMAD2*, in turn, downregulated in response to chemotherapy treatment in breast cancer clinical samples, as indicated in Fig 6.

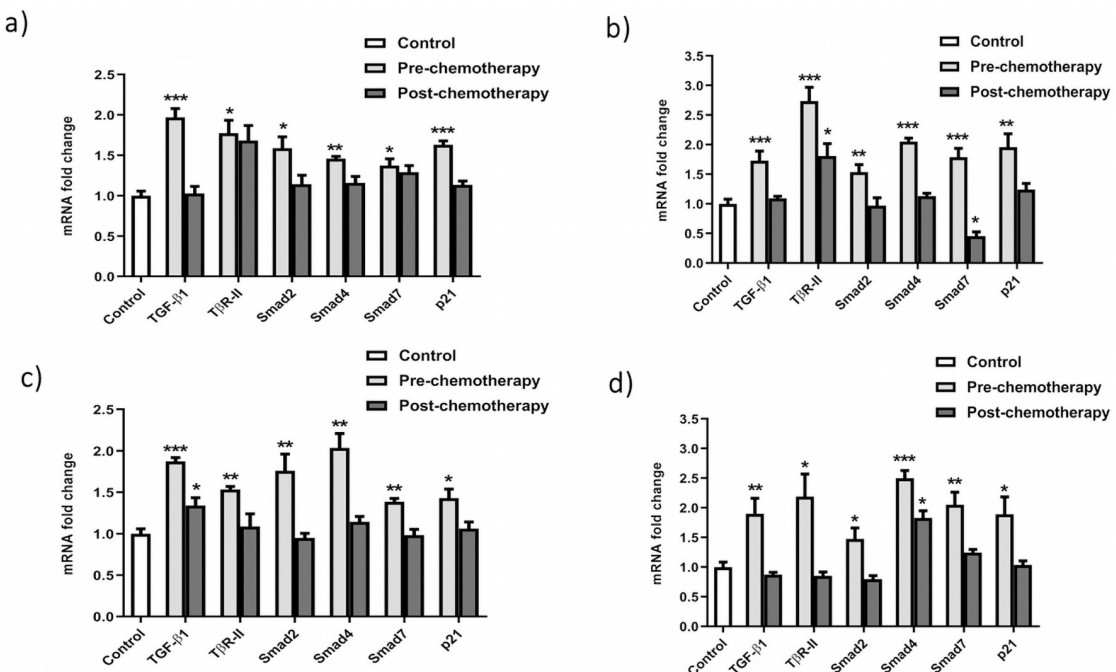

**Fig 4. TGF-β cascade expression profiling in a) G-II blood samples b) G-II biopsy samples c) G-III blood samples d) G-III biopsy samples.** The relative quantification results indicate the expression pattern of *TGF-β* pathway genes in G-II and GIII breast cancer clinical samples based on chemotherapy status. All the experiments were performed in triplicates. Error bars show ±SD. ANOVA test was applied to the data at *P<0.05, **P<0.001, ***P<0.0001.

## Discussion

Breast cancer is the 2nd most common neoplastic malignancy globally, causing mortality after lung cancer in females. Approximately 1.7 million (11.9% of all cancers) new cases of breast cancer are diagnosed every year, and about > 0.45 million people die due to this [31]. There is an alarming increase of 3 to 4% annually in breast cancer in Pakistan compared to a global incidence of 0.5%. It can be controlled by maintaining a population-based registry rather than the hospital or institution-based data, which cannot tell about the precise occurrence statistic in our population. Cancer diagnosis and its treatment has been improved in most advanced countries of the world that decreased the death statistics. However, in less developed countries, there is still a problem of early diagnosis and treatment facilities [32]. In Pakistan, most patients are diagnosed at advanced clinical stage (III or IV) when treatment options are very few, and recovery is primarily impossible; hence, the survival ratio is low, as evident in current research [33]. Awareness, better health standards, and screening for early detection and identification of new tumor prognostic molecular markers can improve the outcome of breast cancer treatment and may also prevent the spread of the tumor into other body parts. Among the clinical prognostic markers, lymph node status, tumor size, and grading system are crucial determining breast cancer progression [33].

The breast tumor grading helps in finding better treatment options for disease control [34]. In the current research, it is observed that invasive ductal carcinoma is more frequent in the Pakistani female population than ductal carcinoma *in situ*. 73.2% patients had tumor grade II, 23.4% grade III, whereas 7 and 2.7% had lump and tumor grade I. Other studies revealed that tumor grade-I and size < 2 cm have an excellent prognostic value compared to high-grade tumors having poor prognosis, more chances of spread of disease with a high mortality rate

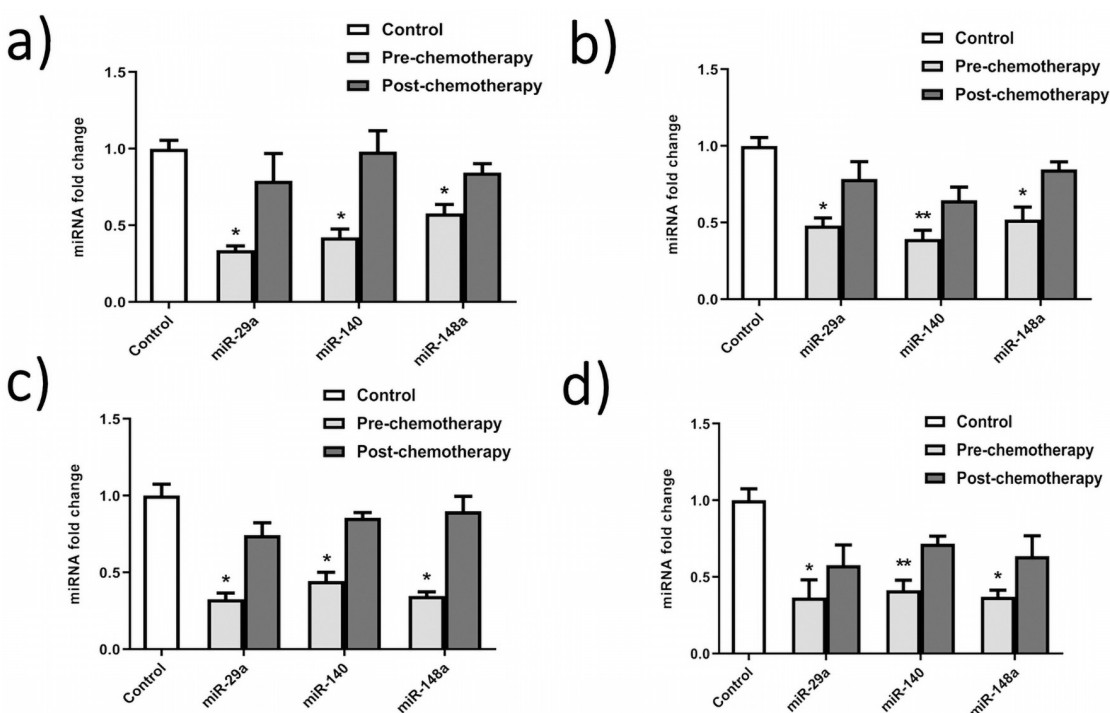

**Fig 5. miRNA expression analysis in breast cancer patients' samples.** a) G-II blood samples b) G-II biopsy samples c) G-III blood samples d) G-III biopsy samples. The qRTPCR results show differential expression patterns of miR-29a, miR-140, miR148 in G-II and GIII breast cancer clinical samples. All the reactions were carried in triplicates. Error bars show ±SD. ANOVA test applied to the data at *P<0.05, **P<0.001.

[35,36]. The factors that are less studied but seem strongly striking to our population include below-average economic status that may link to the psychological effect on the body [37,38]. Stress is one of the less frequently studied risk factors linked with breast cancer. Stress-related hormones cortisol are correlated to this carcinoma because they may suppress the apoptotic and DNA repair activity. Moreover, they affect gene expression profile and exert a negative impact on the body's immune system [38,39]. In the current research, most of the patients observed at extreme stress levels are essentially linked to poor economic status. Herein,

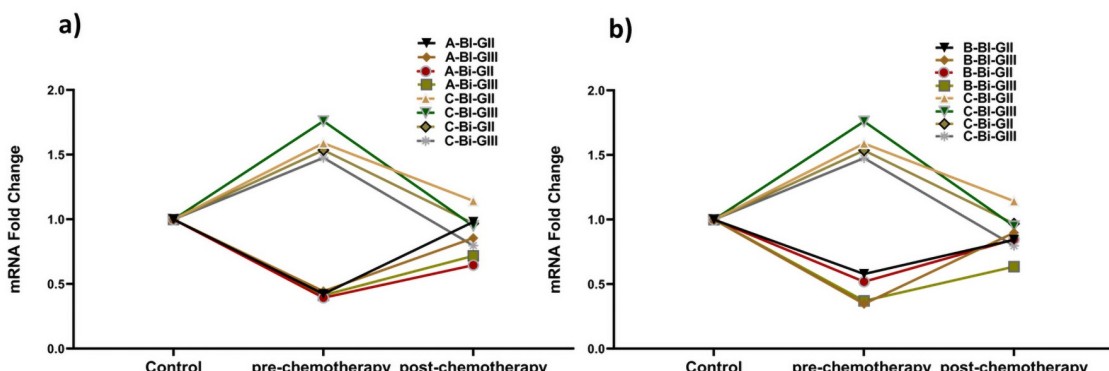

**Fig 6. Correlating expression pattern of miRNAs and *SMAD2* in breast cancer clinical samples.** a) miR-140 b) miR-148a. The results display inverse relation of miRNAs with *SMAD2* as in pre-chemotherapy patients downregulated miRNAs cause *SMAD2* to upregulate in cancer.

patients from the different economic categories, i.e., upper, average, and poor, were studied. These results indicated a link between an upsurge of breast cancer incidence and mortality with poor economic status as reported in India [40]. Thus, there is a dire need to explore this link between poor financial status and stress factors, its duration, and time of exposure to the breast that ultimately changes breast epithelium further in our population. One-third of breast cancer is attributed to established risk factors, while other aspects are yet to be discovered. Therefore, other factors, including environmental factors, might be a strong contributor to breast cancer risk [41–43]. The molecular or genetics of the patients are the other major factor in cancer development. In cancer cells, the different pathways go awry.

*TGF-β1* is the member of the cytokines super-family and *TGF-β1* sub-family. Polymorphism of this gene is reported in Asian populations and is significantly associated with breast cancer risk [44,45]. Furthermore, it correlates with the elevated levels of *TGF-β1* in Indian females [45]. The semi-quantitative and quantitative results presented a 1-fold increase in the expression of *TGF-β1* in both blood and biopsy samples with almost similar levels in different grades. A reduction was seen in patients undergoing chemotherapy, showing a positive response to the treatment. *TGF-β1* acts as a regulator of cell growth by arresting the cells at the G1 phase in normal cells as well as in tumors in the early stages of cancer [46]. On the other hand, at advanced tumor stages, *TGF-β1* shows immunosuppressive activity which promotes growth, angiogenesis, and metastasis of cancer cells. *TGF-β1* also causes the death of the normal cells around the tumor microenvironment and eradicates their antitumor effect [47]. The significant expression level of TGF-β1 revealed its oncogenic activity and suggests that increased expression shifts its activity from growth inhibitor to growth inducer. *TβRII* is the receptor of the *TGF-β1* ligand, and it belongs to the serine-threonine kinase family of receptors. The expression of *TβRII* depicted an uprise of 0.8 to 1.7 times in blood and biopsy samples of G-II patients, respectively. While 0.5 to 1.2-fold increase in G-III patients' blood and biopsy samples, respectively. Its increased expression allows more *TGF-β1* molecules to bind with it and initiate the signal cascade and has been reported to increase the metastasis to lymph nodes [48]. The increased level of *TβRII* along with increased *TGF-β1* expression indicated an up-regulation of the pathway and disclosed its metastatic role in breast cancer. *SMAD2* is a co-receptor and helps in the transduction of signals by making a complex with the *TGF-β1* receptor [49]. Blood samples showed a change of 0.6 folds in the transcript level of *SMAD2* and a similar shift in biopsy samples. Increased *SMAD2* expression has been reported in invasive breast carcinoma, which promotes the signal translocation to the nucleus to promote the expression of target genes [49–51]. The higher level of *SMAD2* can modulate the increased activation of *SMAD4*, which helps *SMAD2* transduce the signal to the nucleus [51].

*SMAD4* is a mediator of downstream signaling of *TGF-β1* [52]. The quantitative results indicated a one-fold change in blood and biopsy samples of patients in both grades. Although its function is unclear, however it is reported to play a role in the spread of cancer to bones. *SMAD4* has also been reported to act as a tumor suppressor in the MDA-MB-468 breast cancer cell line. Here its expression level was seen to be decreased from lower to higher grades [53]. On the other hand, high expression of *SMAD4* in the MDA-MB-231 breast cancer cell line has been seen to increase the basal level of IL-11 on stimulation by *TGF-β1* [53]. As *SMAD4* is directly involved in the transcription of target genes, its higher expression can be related to the invasiveness of breast cancer to other parts like lungs, bones, and axillary lymph nodes [53]. *SMAD7* works as an antagonist of *TGF-β1* signaling [27]. The expression of *SMAD7* is increased in *TGF-β1* signaling as it plays a role in feedback inhibition [27]. The expression of *SMAD7* was significantly increased in G-II patients with a raise of double value than control samples.

The *p21* is a tumor suppressor gene and promoter of apoptosis by CDK inhibition [54]. But it can act both as p53-dependent and p53-independent mechanisms and behave according to cellular context [54]. The transcript level of *p21* was seen to be slightly increased (0.5-fold) in blood, whereas 1-fold in biopsy samples, which suggests that it might play both tumor suppression and tumor induction roles. It can protect the cells from apoptosis even in the up-regulation of *p53* by binding to caspases and inhibiting their activity [55]. The *p21* has also been reported to induce cellular growth while arresting the cell cycle and repairing DNA damage [55]. The oncogenic activity of *p21* in breast cancer has been reported by its inability to inhibit CDKs irrespective of its presence in high levels and by its association with the up-regulation of *HER2/Neu* which results in the poor prognosis of breast cancer [56,57].

The poor prognosis of breast cancer has also been found to be associated with the deregulation of miRNAs. miR-29a acts as a tumor suppressor, and it has been reported to be down-regulated during the development of breast cancer [58]. The expression analysis of miR-29a showed approximately one times less expression in blood and biopsy samples in G-II and G-III patients, respectively. A study in breast cancer cell line MDA-MB-453 revealed that its expression arrests the cells at the G0 or G1 phase of the cell cycle and inhibits its progression by targeting CDC42 [58]. Its slow expression has also been reported to increase drug resistance in breast cancer. miR-140 is known to inhibit the translation of *TGF-β1* and *SMAD3* receptors and observed to be downregulated up to 2-folds in breast cancer [59,60]. The quantitative analysis presented that there was an average drop of one-fold in patients' blood and biopsy samples in G-II and G-III. Loss of miR-140 expression has been shown to increase the stem cell-like properties of primary tumor cells at an early stage of breast cancer [61]. The expression of miR-148a has depicted the inhibition of cancer progression, invasiveness, and migration to other sites [62]. The analysis showed a 1–1.4 times decreased expression of miR-148a, which presented a very significant change in blood and biopsy samples of G-II to G-III patients. Our *in-silico* results predicted *TGFβR* and *SMAD2* gene as potential targets of all three miRNAs. However, validated targets confirmed through several researches revealed an inverse relation of miR-140 and miR-148a with *SMAD2* [63,64] and miR-29a with *SMAD4* gene in different tumors [65]. Interestingly, following the *in-silico* study, a similar pattern was observed in pre-chemo breast cancer clinical samples used in the study. miR-148a has been reported to target the *SMAD2* in the *TGF-β1* signaling pathway to inhibit angiogenesis. Furthermore, It also acts as a promoter of cell death in the MCF-7 breast cancer cell line [62]. Hence, miR-29a, miR-140, and miR-148a seem potential therapeutic and diagnostic candidates in breast cancer management. Moreover, they may pave the way for the development of new and effective therapies for effective disease treatment.

## Conclusions

Cancer is the most challenging disease with a difficult diagnosis and poor prognosis. Invasive Ductal Carcinoma (90%) and grade-II tumor (73.2%) formation are more common in the Pakistani female population. Clinical parameters such as age (31–50 Y), disturbed menstrual cycle ($> 2$), cousin marriages (repeated), and lactation period ($< 0.5$ Y) along with additional factors, i.e., stress, dietary and environmental aspects, have an essential role in the development of breast cancer in females of Pakistani population. Furthermore, the *TGF-β1* signaling pathway and integrated miRNAs have a significant role in breast cancer development. The correlation between blood and biopsy samples could help develop non-invasive diagnostic methods, using the selected genes as biomarkers of breast cancer. As in the current study, *TGF-β1* cascade genes show relatively high expression in pre-chemo samples of both G-II and GIII blood and biopsy samples compared to normal. Moreover, miR-29a, miR-140, and miR-148a

were found directly linked with *SMAD4* and *SMAD2* gene expression. Their low expression level upregulates *SMAD4* and *SMAD2* genes in breast cancer patient samples. Hence, these miRNAs can be presented as a possible therapeutic target to control disease progression. In addition, to find more potential therapeutic targets, the RNA silencing technique can be utilized to knock down or increase the expression of both up regulating and downregulating genes and integrated miRNA, respectively. This may help in rapid diagnosis and improving patient survival.

## Supporting information

**S1 Table. Sequences of *TFGβ* pathway primers.**
(PDF)

**S2 Table. Sequences of miRNA primers.**
(PDF)

**S1 File. English questionnaire for patient history and clinicopathological data.**
(PDF)

**S2 File. Questionnaire for patient history and clinicopathological data in original language.**
(PDF)

## Acknowledgments

We sincerely acknowledge the doctors who aided in the data collection and patients sample collection. We are also thankful to the patients who agreed to participate in the study.

## Author Contributions

**Conceptualization:** Bushra Ijaz.

**Data curation:** Fayyaz Ahmed, Ayesha Malik.

**Formal analysis:** Fayyaz Ahmed, Ayesha Malik, Bushra Ijaz.

**Investigation:** Fayyaz Ahmed, Muhammad Adnan, Somayya Tariq, Farukh Kamal.

**Methodology:** Fayyaz Ahmed, Muhammad Adnan, Ayesha Malik.

**Supervision:** Bushra Ijaz.

**Validation:** Farukh Kamal.

**Visualization:** Farukh Kamal.

**Writing – original draft:** Fayyaz Ahmed, Muhammad Adnan.

**Writing – review & editing:** Somayya Tariq, Bushra Ijaz.

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
