## [Decision Letter · Decision Letter 0]

3 Jun 2021

PONE-D-21-14182

Perception of breast cancer risk factors: Dysregulation of TGF-β/miRNA axis in Pakistani females

PLOS ONE

Dear Dr. Ijaz,

Thank you for submitting your manuscript to PLOS ONE. After careful consideration, we feel that it has merit but does not fully meet PLOS ONE’s publication criteria as it currently stands. Therefore, we invite you to submit a revised version of the manuscript that addresses the points raised during the review process.

Your manuscript has been carefully reviewed and reviewers recommended the minor revisions. Please revise your manuscript according to the reviewers' comments and submit its revised version.

We look forward to receiving your revised manuscript.

Kind regards,

Muhammad Shareef Masoud, Ph.D

Academic Editor

PLOS ONE

Journal Requirements:

Reviewers' comments:

Reviewer's Responses to Questions

**Comments to the Author**

1. Is the manuscript technically sound, and do the data support the conclusions?

Reviewer #1: Yes

Reviewer #2: Yes

2. Has the statistical analysis been performed appropriately and rigorously? 

Reviewer #1: Yes

Reviewer #2: Yes

3. Have the authors made all data underlying the findings in their manuscript fully available?

Reviewer #1: Yes

Reviewer #2: Yes

4. Is the manuscript presented in an intelligible fashion and written in standard English?

Reviewer #1: Yes

Reviewer #2: Yes

5. Review Comments to the Author

Reviewer #1: In the manuscript ‘Perception of breast cancer risk factors: Dysregulation of TGF-β/miRNA axis in Pakistani females’, Fayyaz etal., highlighted the different risk factors like stress, no. of children, menstrual cycles and correlated with age, stage and grade etc. of the diseases. The important thing is they performed in silico and study to pick the miRNA regulating the TGF beta pathway and confirmed the expression in clinical samples both in blood as well as biopsy. The results can help in designing the therapeutic strategy and monitoring the disease state in the population for better output. The manuscript is acceptable for publication after incorporating following revisions.

1. In abstract authors stated that miRNA ‘significantly dysregulated’, it is better to mention either the P value or the fold change observed in the study.

2. In introduction highlight and compare the risk factors of breast cancer observed in current study with that of other Asian country data.

3. In materials and methods section primer sequences used for expression analysis of TGF-beta pathway genes and miRNA are not provided. Supply it as supplementary file.

4. Result should be concluded with ‘May or can’ as more studies are required on large data set to be sure to state.

5. Under heading ‘miRNA-target genes network analysis’, where authors mentioned the ‘different cascades’, it’s better to name few important one.

6. Improve the quality of figure 6.

7. Carefully remove the typo-errors from the manuscript.

Reviewer #2: In the manuscript Fayyaz et al., studied the risk factors of breast cancer and TGF-beta pathway/miRNA interaction in Pakistani population. The result highlights the different factors that can contribute as risk factors of breast cancer. In this study, the clinical parameter, In silico and expression of TGF beta pathway genes and miRNA were studied in clinical samples (both blood and biopsy). Overall, the study is well resented, a few minor changes can be done.

1. The reaction conditions for the synthesis of miRNA cDNA should be given.

2. Where possible provide references for the methods used in the study.

3. In Table 2 authors do not need to mention Pearson correlation with each parameter studied as this information is already provided in title of the Table 2.

4. In Table 2 mention r and p-value with respective values of the correlation study.

5. In result section under heading ‘Demographic Profiling of Breast Cancer Patients’ remove the word elucidation with observed,.

6. In discussion section compare your results with other Asian countries data to highlight the differences if any observed.

7. It is suggested to mention the number of experimental repeats in figure legends.

8. Improve the quality of figures.

9. Remove the typo errors from the manuscript.

6. PLOS authors have the option to publish the peer review history of their article (what does this mean?). If published, this will include your full peer review and any attached files.

Reviewer #1: **Yes: **Sobia Manzoor, PhD

Reviewer #2: **Yes: **Saba Khaliq

---

## [Author Response · Author response to Decision Letter 0]

6 Jul 2021

Editors Comments

Response: Revised manuscript is set according to the PLOS ONE’s style. 

Response: The reference list is thoroughly checked There is no retracted article cited in the manuscript. 

3. Please include additional information regarding the survey or questionnaire used in the study and ensure that you have provided sufficient details that others could replicate the analyses. For instance, if you developed a questionnaire as part of this study and it is not under a copyright more restrictive than CC-BY, please include a copy, in both the original language and English, as Supporting Information 

Response: The questionnaire used in the study is provided as supplementary file S1 (English) and S2 (in original Language). Please see the S1 and S2 files

REVIEWER 1

1. In abstract authors stated that miRNA ‘significantly dysregulated’, it is better to mention either the P value or the fold change observed in the study. 

Response: Thank you, reviewer, for the comment. The P value has been added to indicate significant change in the abstract. Please see the revised manuscript abstract, addition is highlighted in green. Please see page No. 2. Line # 41.

2. In introduction highlight and compare the risk factors of breast cancer observed in current study with that of other Asian country data. 

Response: The comparison is added in the revised manuscript introduction highlighted in green. Please see page No. 3, line # 64-67,Ref 7-10.

 3. In materials and methods section primer sequences used for expression analysis of TGF-beta pathway genes and miRNA are not provided. Supply it as supplementary file. 

Response: The primer sequences have been provided as supplementary file S1 and S2 Tables in revised manuscript Please see the S1 and S2 Tables

4. Result should be concluded with ‘May or can’ as more studies are required on large data set to be sure to state. 

Response: The results have been concluded with either may or can. Please see the revised manuscript the changes have been highlighted in green. Page No. 2, 22, and 23, 25-27. Line # 45, 370, 380, 383, 429, 484 and 487.

5. Under heading ‘miRNA-target genes network analysis’, where authors mentioned the ‘different cascades’, it’s better to name few important one. 

Response: The miRNA target cascades have been added under the heading ‘miRNA-target genes network analysis’. Please see the revised manuscript the changes have been highlighted in green. Page NO.17, line # 255-257.

6. Improve the quality of figure 6.

 Response: The quality of figure 6 has been improved in revised manuscript. Please see figure 6.

7. Carefully remove the typo-errors from the manuscript.

Response: The revised manuscript has been thoroughly checked and edit to remove the typo-errors 

REVIEWER 2

1. The reaction conditions for the synthesis of miRNA cDNA should be given. 

Response: The reaction condition has been added in the revised manuscript. Please see the heading synthesis of miRNA cDNA. The addition is highlighted in yellow, Page No. 9, Line # 174-177.

2. Where possible provide references for the methods used in the study. 

Response: References has been added in the revised manuscript materials and methods highlighted in yellow. Please see page No. 8 line # 163 and Page # 9, line No. 184.

3. In Table 2 authors do not need to mention Pearson correlation with each parameter studied as this information is already provided in title of the Table 2. 

Response: Table 2 has been revised as suggested by the reviewer. The changes have been highlighted in yellow. Please see Table 2 at page No.15 & 16. 

4. In Table 2 mention r and p-value with respective values of the correlation study. 

Response: Table 2 has been revised as suggested by the reviewer. The changes have been highlighted in yellow. Please see Table 2 at page No.15 & 16.

5. In result section under heading ‘Demographic Profiling of Breast Cancer Patients’ remove the word elucidation with observed.

Response: The word elucidation is replaced with observed. This change has been highlighted in yellow in the revised manuscript. Please see page No. 14. Line #238

6. In discussion section compare your results with other Asian countries data to highlight the differences if any observed.

Response: The information is added in the revised manuscript. The addition is highlighted in yellow in revised manuscript. Please see page No. 23. Line # 388, 397-399. And Ref No. 40, 44 and 45.

7. It is suggested to mention the number of experimental repeats in figure legends. 

Response: The number of experimental repeats has been added in the revised manuscript, highlighted in yellow. Please see figure 4 legend at page No. 20, line # 320 and Figure 5 legend at page No. 21 line # 337.

8. Improve the quality of figures. 

Response: Quality of Figures have been improved in the revised manuscript. 

9. Remove the typo errors from the manuscript. 

Response: The typo-errors have been removed from the revised manuscript.

---

## [Editor Report · Decision Letter 1]

13 Jul 2021

Perception of breast cancer risk factors: Dysregulation of TGF-β/miRNA axis in Pakistani females

PONE-D-21-14182R1

Dear Dr. Ijaz,

We’re pleased to inform you that your manuscript has been judged scientifically suitable for publication and will be formally accepted for publication once it meets all outstanding technical requirements.

Kind regards,

Muhammad Shareef Masoud, Ph.D

Academic Editor

PLOS ONE
---

## [Editor Report · Acceptance letter]

16 Jul 2021

PONE-D-21-14182R1 

Perception of breast cancer risk factors: Dysregulation of TGF-β/miRNA axis in Pakistani females 

Dear Dr. Ijaz:

I'm pleased to inform you that your manuscript has been deemed suitable for publication in PLOS ONE. Congratulations! Your manuscript is now with our production department. 

Kind regards, 

on behalf of

Dr. Muhammad Shareef Masoud 

Academic Editor

PLOS ONE